# ATTRIBUTE-CENTRIC REPRESENTATION LEARNING FOR INTERPRETABLE CRIME SCENE ANALYSIS IN VIDEO ANOMALY DETECTION

## ABSTRACT

Automatic crime scene analysis is an important application area for representation learning in Video Anomaly Detection (VAD). Effective interpretation of anomalous events requires models to learn rich, disentangled representations that capture fine-grained, crime-relevant attributes. However, widely used VAD datasets (e.g., UCA, CUVA) primarily offer coarse event-level labels and they lack attribute-level supervision often needed for modeling crime-specific behaviors. To bridge this gap, we propose an attribute-centric learning framework that explicitly conditions video representations on crime-causing attributes. We extend the UCA dataset with over 1.5M new attribute-centric annotations generated using carefully designed prompts and LLMs. These annotations enable supervised fine-tuning of a curated CLIP-based model, leading to more discriminative, attribute-aware video representations, and precise event captions. An LLM-based summarizer then distills these captions into context-rich explanations, facilitating interpretable scene understanding. Our approach answers three core questions in crime scene analysis: **What? When? How?** Extensive experiments show that the proposed representation learning framework yields significant improvements ($\approx 20\% \uparrow$) in attribute-centric crime classification accuracy and ($\approx 6.4\% \uparrow$) according to MMEval scores over the baselines. We further analyze and mitigate biases in MMEval to ensure robustness and fair evaluation. These results highlight the importance of attribute-conditioned representation learning for interpretable and reliable VAD.

## 1 INTRODUCTION

Crime scene analysis using video anomaly detection (VAD) is aimed at indexing and describing abnormal situations that may arise from unusual activities of actors. Law enforcing agencies can take its help to identify and localize potential crime incidents through visual surveillance Sultani et al. (2018); Lu et al. (2013); Wu et al. (2022b); Mahadevan et al. (2010). However, existing VAD methods Wu et al. (2023); Del et al. (2021) face significant challenges. These methods require (i) extensive human-annotated data Yuan et al. (2024); Du et al. (2024), making them difficult to use for large-scale applications, (ii) they primarily focus on coarse-grained anomaly detection, failing to provide interpretable, attribute-rich descriptions of crime events, and (iii) certain anomalies do not cause sudden scene change, making them difficult to be detected using traditional VAD techniques Wu et al. (2023); Zhang et al. (2023); Chang & Wang (2023); Joo et al. (2023).

Assume a shooting incident is happening at a public place. The act of firing a gun may not always cause a significant change in the scene. An arson act may begin with inconspicuous action, such as pouring accelerant or positioning materials. Detecting such events are challenging using existing VAD models due to the subtle nature and similarity of such events with non-anomalous traits. In case of vandalism, the damage may be gradual and less obvious unless accompanied by sudden physical destruction Sabokrou et al. (2018). These examples highlight the need for designing VAD models that can capture underlying, often subtle, cues or attributes associated with the underlying behavior to ensure a comprehensive monitoring. Datasets like UCA Yuan et al. (2024) and CUVA Du et al. (2024) play important roles in training the VLMs to describe anomaly events. However, generating fine-grained descriptions by leveraging crime-specific attributes, has not yet been

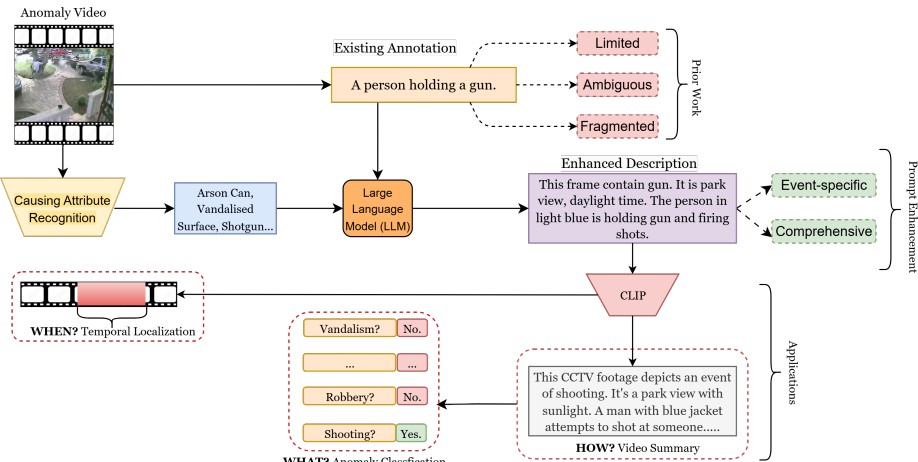

Figure 1: **Attribute-centric Learning:** In CUVA Du et al. (2024) dataset, annotations are tasks specific (focus on broad tasks ignoring subtle and critical attributions). UCA Yuan et al. (2024) dataset's annotations are limited, short-sentenced and provide limited interpretation, hindering accurate detection and event description. The proposed method tackles this by first detecting crime-specific attribute-centric learning, such as *firearms*. Such learning enriches the annotations with this information, and leverage LLM inference to generate detailed and robust crime scene descriptions.

explored. We leverage the advantages of existing datasets and provide an attribute-centric learning framework for crime scene analysis. It has the following advantages: (i) We are able to generate attribute-rich, context-sensitive captions using a CLIP-based model and extend it for summarization using LLMs. (ii) Temporal localization has been enhanced by identifying key-event frames using **similarity scores** for a precise (frame-level) classification. (iii) The attribute-centric learning framework has improved crime categorization accuracy by integrating attribute-based features. Overall, the approach as depicted in Figure 1, has led us to answer the following important questions: **What, When, and How** an anomaly is happening?

UCF Crime dataset Sultani et al. (2018) contains 13 classes of anomalies. Certain categories such as *arson*, *vandalism*, or *shooting* can be described in details using a well-defined set of "crime-causing attributes". This makes attribute-centric learning feasible for these categories as compared to other classes. For example, objects like lighters, matchsticks, or gasoline cans are indicative of arson, firearms are necessary for shooting, and special tools are needed for vandalism. A few types of crimes lack distinguishable attributes. For example, a robbery incident does not mandatorily need a crime causing attribute. Despite this, it is possible to design a generic method that works reasonably well across all classes, though the effect of "crime-causing attributes" will be limited on certain classes. To achieve this, we extend the UCA (UCF Crime with Attributes) dataset Yuan et al. (2024) by integrating detailed attribute annotations for these crimes. Unlike traditional VAD datasets, which lack attribute-specific information Luo et al. (2017); Wu et al. (2022b), the augmentation improves descriptive granularity for crime scene analysis. We leverage DeepMAR Li et al. (2018), a popular Person Attribute Recognition (PAR) model trained on a custom dataset to recognize relevant crime attributes. This approach enhances scene understanding by focusing on causation-aware features, allowing for early detection of criminal activity. For instance, detecting a person handling flammable materials can serve as an early indicator of arson, even before any fire is visible. In summary, our focus is on attribute-centric learning for crime scene analysis by analyzing the presence of distinguishable crime-causing attributes. To accomplish this, we have made the following technical contributions:

- **Attribute-enriched annotations:** We extend the event descriptions available in the UCA dataset by incorporating crime-specific attributes, enabling detailed and interpretable anomaly detection.

- **What?** We categorize the crimes among various types, leveraging attribute cues to determine the specific type in a given video.

- **When?** We present a new CLIP-based attribute-centric learning framework for crime frame localization to identify the frames containing the crime incident for a precise temporal analysis.

- **How?** We introduce attribute-centric learning method to improve the summarization capability for generating focused summary of crime-related incidents.

The remainder of this paper is structured as follows: Prior works are presented in the next section. We describe the proposed methodology in the following section followed by experiments and results. Finally, we conclude with a few future directions.

## 2 PRIOR WORK

Popular anomaly detection works Sultani et al. (2018); Wu et al. (2022b); Majhi et al. (2021); Tian et al. (2021); Li et al. (2022b); Feng et al. (2021); Leroux et al. (2022); Pang et al. (2020); Li et al. (2022a); Wu et al. (2022a); Luo et al. (2017); Acsintoae et al. (2022); Liu & Ma (2019); Thakare et al. (2023) formalize VAD as a binary classification task, leveraging a ranking loss function to distinguish between normal and abnormal events. Though effective, this approach has limitations, especially in scenarios, where criminal actions involve subtle and gradual changes that may not immediately trigger a high anomaly score.

**CLIP-based Anomaly Detection and Localization.** More recent advancements in VAD, such as VAD-CLIP Wu et al. (2023), leverage vision-language models like CLIP Radford et al. (2021) and calculate the cosine similarity between video frames and descriptive textual prompts. Zhang *et al.* Zhang et al. (2023) have introduced a CLIP-guided visual-text fusion transformer, merging textual descriptions with visual data to improve attribute recognition, an essential component for understanding nuanced crime scenarios. Similarly, Chang *et al.* Chang & Wang (2023) have demonstrated the efficiency of combining CLIP embeddings with temporal attention mechanisms to better capture frame-level anomaly patterns. Expanding on CLIP-based approaches, Joo *et al.* Joo et al. (2023) have proposed CLIP-TSA. It incorporates a temporal self-attention mechanism to enhance weakly-supervised video anomaly detection. This method effectively captures temporal dependencies and improves the model's ability to identify anomalous events. Zanella *et al.* Zanella et al. (2023) have explored the CLIP latent space for video anomaly recognition, demonstrating that carefully designed prompts and feature extraction techniques can significantly boost performance in identifying abnormal events. Kumar *et al.* Kumar & Singh (2024) have explored a hybrid CLIP model to detect human-object interactions. It is useful for identifying high-risk actions within a sequence. Despite these advances, these methods lack temporal sensitivity to handle complex causative attributes attached to crimes.

**Causation in VAD.** Recent works have introduced causative factors in VAD. Wu *et al.* Du et al. (2024) have proposed a benchmark that combines visual and causal cues to predict not only *what* happened but also *why* and *how*. Such causal insights enable models to detect anomalies even when visual deviations are subtle but causal indicators suggest criminal activity. Another notable work by Li *et al.* Li & Zhou (2024) introduces context-aware causal cues that further enhances localization accuracy by detecting scene context changes indicative of suspicious behavior. While promising, these methods do not yet address causative attributes specific to different crime types, such as "handling flammable materials" or "concealing a weapon," which are essential for proactive detection.

**Training-free Approaches.** Exploration of training-free methods Zanella et al. (2024); Gu et al. (2024) are recently introduced in VAD. Zanella *et al.* Zanella et al. (2024) have introduced a novel approach that harnesses LLMs for video anomaly detection without requiring task-specific training. While these approaches offer flexibility, they remain limited in fine-grained crime detection due to the lack of specialized attribute conditioning.

**LLM-assisted Anomaly Detection.** LLMs have also been used to enhance anomaly detection and explanation. Holmes-VAD Zhang et al. (2024) proposes an LLM-guided framework to achieve unbiased and explainable VAD. Similarly, AnomalyRuler Yang et al. (2024) introduces a rule-based reasoning framework for VAD with LLMs. Additionally, HAWK Tang et al. (2024) leverages interactive large VLMs to interpret video anomalies precisely, integrating motion modality to enhance

anomaly identification and constructing an auxiliary consistency loss to guide the video branch to focus on motion modalities.

**Datasets and Limitations.** UCA Yuan et al. (2024) stands out for its focus on criminal activities. It further enhances crime understanding by offering sentence-level prompts, yet it is limited to general descriptors without attribute-specific granularity needed for precise crime analysis. XD-Violence Wu et al. (2022b), another critical dataset, extends the scope of anomaly detection by covering violent events. However, it lacks sufficient crime-specific labels, limiting its application to broader VAD contexts rather than targeted crime analysis. CUHK Avenue Lu et al. (2013) and UCSD Pedestrian (Ped1 and Ped2) Mahadevan et al. (2010), both designed for pedestrian anomaly detection, provide bounding-box annotations but lack the complexity of diverse crime scenarios, making them less applicable to nuanced crime detection tasks. While these datasets contribute to anomaly understanding, they exhibit limitations in providing fine-grained temporal, contextual, and causative annotations necessary for crime-specific VAD tasks.

Our approach addresses the aforementioned gaps in datasets as well as CLIP-based models by introducing causative prompts tailored to high-impact crimes. By combining these cues, the proposed framework enables frame-level localization and enhanced scene descriptions tailored to different crime types, achieving improved performance when compared to existing VAD methods. In summary, by incorporating both visual and causative features, we build on recent advancements in causal VAD to provide a more holistic understanding of crime scenes by answering the questions: **What, When, and How?**

## 3 PROPOSED METHOD

### 3.1 PROBLEM FORMULATION

VAD tasks focus on detecting deviations from normal activity patterns in surveillance videos. Integrating textual descriptions with visual features for crime scene analysis has gained traction Du et al. (2024)Yuan et al. (2024). CLIP models Radford et al. (2021); Zhang et al. (2023) have been utilized for such tasks Wu et al. (2023), where the similarity score $s$ is computed as shown in Eq. (1),

$$s = \frac{\mathbf{v} \cdot \mathbf{t}}{\|\mathbf{v}\|\|\mathbf{t}\|} \tag{1}$$

where $\mathbf{v}$ is video frame embedding and $\mathbf{t}$ is the textual prompt embedding. However, CLIP struggles with context-dependent anomalies like gradual vandalism. For instance, when "a person starts spray painting and defaces the wall over time," early frames might not seem anomalous, but cumulatively, it becomes an act of vandalism. Preparatory actions similarly fail to yield high similarity scores via standard visual-textual matching. Moreover, preparatory actions may not yield high similarity scores through the visual-textual approach alone. To address this, we propose integrating causative attributes. We redefine the anomaly score as shown in Eq. (2),

$$(x_i) = \alpha f(x_i) + \beta g(\mathbf{c}_i) \tag{2}$$

where $f(x_i)$ captures the visual anomaly score, $g(\mathbf{c}_i)$ encapsulates causative attribute information, and $\alpha, \beta$ are weighting factors.

For scenarios where crime-specific attributes are available, we propose a joint embedding space by projecting video frames $\mathbf{v}_i$ and textual prompts $\mathbf{t}_j$ using projection matrices $\mathbf{W}_v$ and $\mathbf{W}_t$, respectively. The similarity score is then computed as $s_{ij} = \cos(\mathbf{W}_v\mathbf{v}_i, \mathbf{W}_t\mathbf{t}_j)$. In this work, the UCA dataset is enriched with curated prompts refined by LLM, integrating crime-specific causative attributes into the CLIP-based framework. The final anomaly score thus becomes $s_i = \gamma \cos(\mathbf{v}_i, \mathbf{t}_{c,i}) + \delta h(\mathbf{c}_i)$, where $\mathbf{t}_{c,i}$ denotes crime-specific textual attributes, $h(\mathbf{c}_i)$ captures causal cues, and $\gamma, \delta$ are weighting factors.

### 3.2 DATASET AUGMENTATION

The dataset augmentation works as follows. Let $\mathcal{D} = \{X_i, Y_i\}_{i=1}^N$ denotes the dataset (UCA), where each video frame $X_i$ is linked to a class label $Y_i \in \{y_{\text{arson}}, y_{\text{shooting}}, y_{\text{vandalism}}, y_{\text{accident},...}\}$. For enhancement with attribute-level information, we establish an attribute set $\mathcal{A} = \{a_1, a_2, \ldots, a_K\}$, where each attribute $a_k$ denotes a crime-related trait such as "firearm possession," "arson tools," or

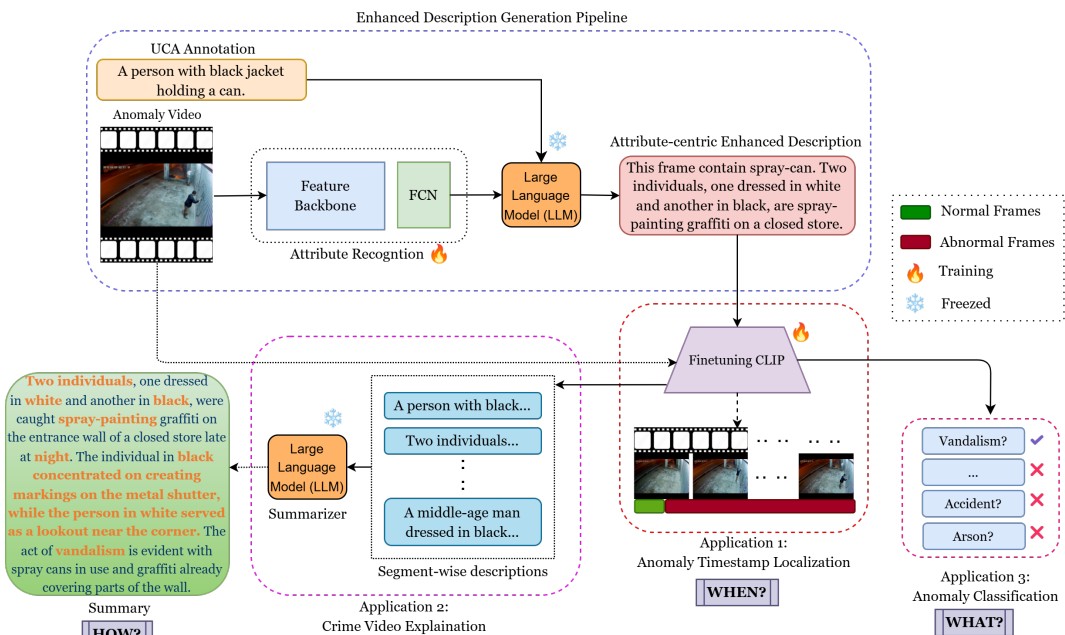

Figure 2: Detected anomaly-causing attributes are fed to a pre-trained LLM for enhancing the anomaly descriptions (upper blue block). Two applications of the proposed pipeline are: (i) fine-tuning a CLIP model with enhanced captions to detect anomalies based on the highest similarity score between anomalous frames and corresponding captions, and (ii) generating segment-wise eventful captions, combining them to produce a final, concise description.

"violent behaviour." Our objective is to map each frame $X_i$ with a corresponding attribute vector $\mathbf{A}_i = [a_{i1}, a_{i2}, \ldots, a_{iK}]$ that denotes the presence or absence of particular attributes. The crime attribute recognition model functions as a multi-label classifier. For each frame $X_i$, an attribute vector $\mathbf{A}_i = [a_{i1}, a_{i2}, \ldots, a_{iK}]$ is predicted using a multi-label classifier (DeepMAR Li et al. (2018)), where the probability of the presence of an attribute is estimated by: $p_{ik} = \sigma(\mathbf{w}_k^\top \phi(X_i) + b_k)$, with $\phi(X_i)$ as the frame's feature representation, $\mathbf{w}_k$ as the attribute weight vector, $b_k$ as bias, and $\sigma(\cdot)$ be the sigmoid function.

It enables the model to recognize multiple attributes per frame, enhancing the accuracy in anomaly descriptions. Post attribute extraction, the initial UCA prompt $P_i$ is enhanced into a detailed prompt $\tilde{P}_i$ by passing $(P_i, \mathbf{A}_i)$ to an LLM, usually, ChatGPT or Gemini OpenAI (2023); Research (2024) are used as the gold standard LLMs. They enhance $P_i$ by incorporating attribute details, formulated as $\tilde{P}_i = \text{LLM}(P_i, \mathbf{A}_i)$. This augmentation enriches the dataset into $\tilde{\mathcal{D}} = \{X_i, Y_i, \tilde{P}_i\}_{i=1}^N$, enabling context-sensitive anomaly descriptions that significantly boost the efficacy of CLIP-based models. By integrating attribute vectors $\mathbf{A}_i$ into the prompts, the proposed framework significantly enhances the descriptive richness of the dataset, which is crucial for training the CLIP model to produce context-sensitive captions. The augmented dataset ($\tilde{\mathcal{D}}$) enables training a model to produce detailed, attribute-aware anomaly descriptions.

## 3.3 CRIME ANALYSIS FRAMEWORK

The proposed framework, as shown in Figure 2, employs the enhanced UCA dataset to train a CLIP model, establishing two core applications: (i) generating detailed video descriptions that provide deeper insights into crime scenes (**How?**), (ii) identifying specific frames where a typical criminal activity occurs (**When?**), and (iii) categorization of the crime (**What?**). Each application is detailed below.

### 3.3.1 CRIME VIDEO DESCRIPTION (HOW?)

The proposed framework generates detailed video descriptions to improve interpretability. The process involves generating prompts specific to extracted attributes, selecting the prompts above a pre-

defined score, and compiling them to produce a cohesive video summary. **Prompt Generation and Similarity Scoring.** For each video, we extract a set of attributes $\mathcal{A} = a_1, a_2, \ldots, a_K$ using an existing causal attribute recognition method Li & Zhou (2024). Each attribute is then processed by an LLM to generate descriptive prompts. For an attribute $a_k$, we formulate a descriptive prompt $p_k = \text{LLM}(\text{"Describe the scene involving } a_k\text{"})$, where the LLM produces a contextually tailored description focused on the specified attribute. Each prompt $p_k$ is paired with frames $X_i$ and scored using CLIP similarity $s(X_i, p_k) = \text{CLIP}(X_i, p_k)$. To filter the most relevant descriptions, prompts with similarity scores above a threshold $\tau$ (i.e., $s(X_i, p_k) \geq \tau$) are selected to form a set $\mathcal{P}$. **Summary Generation with LLM.** The selected prompts, $\mathcal{P} = p_k \mid s(X_i, p_k) \geq \tau$, are then passed to a summarization model, which integrates them into a cohesive video description. The summarizer, powered by an LLM, produces a comprehensive summary $\tilde{S} = \text{Summarizer}(\mathcal{P})$ of the input video, where $\tilde{S}$ encapsulates a structured narrative of the crime event captured across the frames.

### 3.3.2 CRIME LOCALIZATION (WHEN?)

To identify the frame where a crime activity is taking place, we utilize crime-specific attributes obtained from the PAR Li et al. (2018) as a pre-processing step. Let $\mathcal{A} = a_1, a_2, \ldots, a_K$ represents the set of attributes associated with relevant crime indicators, such as "firearm possession", "arson tools", etc. Each attribute $a_k$ is then mapped to a structured prompt $p_k$ in the form $p_k = $ "This frame contains $a_k$". **Similarity Matching with CLIP.** Upon generating the prompts $(p_k)$, we input them along with the video frames into the trained CLIP model. Using CLIP, we compute similarity scores as given in Eq. (3), for every frame $X_i$.

$$s(X_i, p_k) = \text{CLIP}(X_i, p_k) \tag{3}$$

The top-5 frames $X^*$ with the highest similarity scores within a video segment $\mathcal{D}_j$ are identified using Eq. (4). This allows us to precisely localize crime actions in terms of their start and end frames.

$$X^* = \arg \max_{X_i \in \mathcal{D}_j} s(X_i, p_k) \tag{4}$$

### 3.3.3 CRIME CATEGORIZATION (WHAT?)

The aim of this application is to categorize a crime in a video segment by identifying crime-relevant attributes in individual frames and mapping them to specific crime categories e.g., arson, shooting, etc. For categorization, we compute the similarity score between each frame $f_j$ and a set of attribute-derived prompts $P_i$, selecting the prompt with the highest similarity as $P_{i^*} = \arg \max_{P_i \in P} S(f_j, P_i)$. **Crime Mapping.** This calculation yields the prompt with the highest similarity score for each frame, and we track the highest overall similarity score across all frames in the video. We define a mapping function $C : A \rightarrow$ Crime Categories, where each attribute $a_i$ is associated with a specific crime category, e.g., arson, shooting, vandalism, etc. Based on the attribute $a_{i^*}$ corresponding to the most similar prompt $P_{i^*}$, we assign the crime category $c$ using $c = C(a_{i^*})$, where $C$ is the attribute-to-crime mapping function.

## 4 EXPERIMENTS AND RESULTS

### 4.1 IMPLEMENTATION DETAILS

We use GPT Achiam et al. (2023) to adopt CLIP-L/14 visual encoder and text encoder to find similarity scores. For attribute recognition, we employ DeepMAR Li et al. (2018), which was pre-trained and fine-tuned on the PA-100K dataset for person attribute recognition and a custom subset of UCA attribute annotations for domain adaptation. Details on this process and data splits are provided in the supplementary. The CLIP model computes cosine similarity between normalized visual and textual embeddings, which naturally ranges from $[-1, 1]$. For all thresholding and localization steps, *we rescale* this value to $[0, 1]$ via $s_{\text{norm}} = (s_{\text{cos}} + 1)/2$ for interpretability and score fusion. Further details about implementation are available in the **supplementary document**.

### 4.2 EVALUATION PROTOCOL AND METRICS

**MMEval Score.** To evaluate the quality of generated textual descriptions, we use the MMEval metric as proposed in Du et al. (2024), which measures descriptive and semantic alignment between

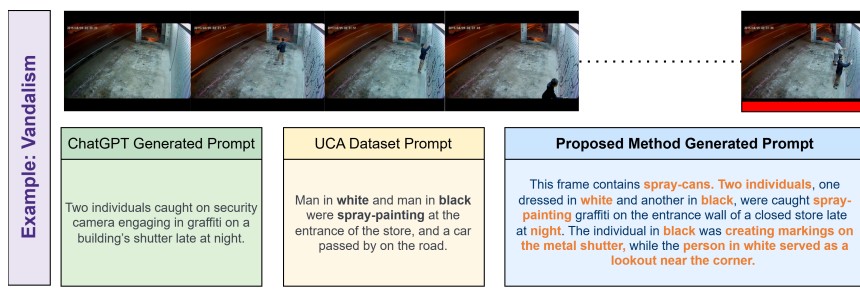

Figure 3: Comparisons of different captioning methods for the frame highlighted in red bar in example: the initial ChatGPT prompt, the UCA dataset prompt, and the proposed prompt enhancement. **The caption is less than 77 tokens in length.** The enhanced prompt offers more comprehensive descriptions of the anomalies, highlighting key elements that improve anomaly detection. Additional results and analysis are available in the **supplementary document**.

| Dataset | Type (Tasks) | Frames | Attributes | Prompts |
|---|---|---|---|---|
| UCF-Crime Sultani et al. (2018) | Crime (13) | 13,741,393 | No | NA |
| XD-Violence Wu et al. (2022b) | Violence (6) | 114,096 | Yes | NA |
| UCSDPed1-Ped2 Mahadevan et al. (2010) | Pedestrian (5) | 14,000 | No | NA |
| CUHKAvenue Lu et al. (2013) | Pedestrian (5) | 30,652 | No | NA |
| UCA Yuan et al. (2024) | Crime (13) | 13,741,393 | Yes | Sentence-level |
| CUVA Du et al. (2024) | Multiple (42) | 3,345,097 | Yes | Free-text |
| **Improved UCA** | **Crime (13)** | **13,741,393** | **Yes** | **Attribute-level** |

Table 1: Comparative analysis of the enhanced UCA dataset with other VAU datasets.

| Captioning Method | MMEval (Score%) |
|---|---|
| ChatGPT Initial Prompt | 70.5 |
| UCA Prompt | 75.3 |
| **Enhanced UCA Prompt** | **86.4** |

Table 2: MMEval scores of various captioning methods.

generated captions and ground-truth or human annotations using a large vision language model. Specifically, for each event, both candidate and reference captions are fed to a state-of-the-art multimodal LLM (e.g., GPT-4V), which assigns a score reflecting the semantic fidelity and detail of the description. While this allows for comprehensive semantic comparison, the metric's reliance on LLMs may introduce a bias favoring longer, more verbose captions. To mitigate confounding factors, we explicitly analyze the effect of caption length on scores in Sec. 4.2.

**Caption Length Bias Analysis.** LLM-based evaluators may reward longer outputs. To quantify this effect, we compute the Pearson correlation coefficient between the number of tokens in captions and their respective MMEval scores across the benchmarks. We find a mild correlation ($r = 0.19$), but this does not fully account for the large gains in MMEval across methods. The observed improvements are primarily due to richer, more relevant attribute content. Furthermore, for fair comparisons, all captions are length-capped at 77 tokens for evaluation.

**Dataset Usage.** We evaluate on UCA Yuan et al. (2024) (attribute-augmented and baseline), UCF-Crime Sultani et al. (2018), and XD-Violence Wu et al. (2022b). For fair comparison, all methods have been trained and evaluated with the same data splits.

### 4.3 BENCHMARKING UCA & DATASET COMPARISONS

In Figure 3, we compare different captioning approaches for describing a graffiti vandalism incident captured by a surveillance camera. The three captions illustrate different levels of descriptive detail: the initial GPT-generated prompt, the UCA dataset prompt, and the proposed prompt enhancement. Each prompt aims to describe the actions and context in the scene, and these differences significantly impact model performance on anomaly detection tasks. The GPT-generated prompts provide a brief and general description, e.g. *"Two individuals caught on security camera engaging in graffiti on a building's shutter late at night"*. Even though it identifies the core activity, the description lacks detail about the individuals' clothing, roles, and surroundings, limiting its utility for context-sensitive anomaly detection. The UCA dataset prompts are slightly more descriptive, noting, *"Men in white and men in black were spray-painting at the entrance of the store, and a car passed by on the road"*. While it highlights the presence of the car and clothing colors, still falls short in specifying the roles or behaviors of each individual. The enhancement of prompt by the proposed method does a better job. It integrates additional attributes and offers a more comprehensive description: *"Two individu-*

*als, one dressed in white and another in black, were caught spray-painting graffiti on the entrance wall of a closed store late at night. The individual in black concentrated on creating markings on the metal shutter, while the person in white served as a lookout near the corner. The act of vandalism is evident with spray cans in use and graffiti already covering parts of the wall"*. This version of the prompt not only captures the appearance and actions of each individual but also emphasizes the coordinated roles in the vandalism, enriching the contextual information available for anomaly detection.

As presented in Table 2, the enhanced captions significantly improve a model's performance in terms of MMEval scores and anomaly precision. The enhanced captions allow the model to better understand complex scenes by distinguishing individual actions and contextual details. This improvement highlights the importance of integrating attribute-level descriptions in captions for robust anomaly detection models. The results emphasize that using detailed annotations leads to more effective detection and interpretation of nuanced actions within surveillance footage.

## 4.4 ABLATION STUDY

In ablation study, we have evaluated the effectiveness of different components in the proposed framework. We examine (i) the performance of large language models (LLMs) for frame description, (ii) the comparative impact of the UCF Crime, UCA, and the proposed augmented datasets, (iii) the influence of attribute-based enhancement on CLIP, (iv) improvements in scene description quality, and (v) localization accuracy across different crime frame localization methods.

### 4.4.1 PROMPT GENERATION COMPARISONS

We have evaluated various LLMs, including GPT-4 Achiam et al. (2023), Gemini, Bard, and Llama, to determine the best-performing model for generating frame descriptions. Table 3 presents the MMEval Du et al. (2024) scores for each LLM, showing that GPT-4 achieves the highest overall performance, closely followed by Gemini Research (2024).

| LLM Model | MMEval Score (%) |
|---|---|
| Llama 3.1 (8B) Touvron et al. (2024) | 76.9 |
| Claude 3.5 Anthropic (2024) | 78.6 |
| Gemini 1.5 Research (2024) | 82.3 |
| **GPT 4.0** Achiam et al. (2023) | **84.1** |

| Crime Category | Subcategories | Accuracy (%) |
|---|---|---|
| Robbery Related | Burglary, Shoplifting, Stealing, Robbery | 32.11 (↓) |
| Explosion | - | 51.45 (↓) |
| Arson | - | **92.38** (↑) |
| Shooting | - | 86.50 (↑) |
| Vandalism | - | 88.41 (↑) |
| **Mean Accuracy** | - | **70.17** (↑) |

Table 3: MMEval scores for different LLMs applied for testing prompt generation performance.

Table 4: Accuracy trends across various crime-related anomalies as compared to Multi-Stream VAD Thakare et al. (2022).

### 4.4.2 EFFECT OF ATTRIBUTE ENHANCEMENT

To demonstrate the effect of attribute-enhancement, we have evaluated CLIP Radford et al. (2021) performance on the proposed enhancement of UCA dataset with and without attribute vectors gen-

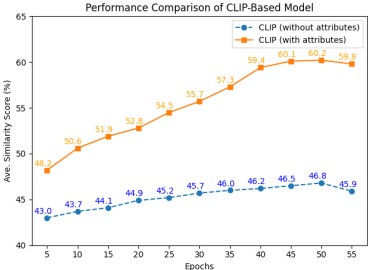

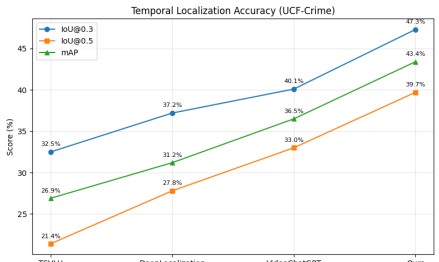

Figure 4: Performance comparisons of CLIP Radford et al. (2021) model with and without the attribute enhancement.

Figure 5: **Temporal Localization Accuracy (UCF-Crime):** Comparison of IoU@0.3, IoU@0.5, and mAP across methods.

erated by DeepMAR Li et al. (2018). As shown in Figure 4, the attribute-enhanced captions yield significant accuracy gain over multiple epochs, indicating that attribute-level details improve model interpretability and context sensitivity.

### 4.5 Application Experiments

#### 4.5.1 Scene Descriptions (How?)

We have conducted a comparative evaluation of the quality of scene descriptions generated by the proposed dataset enhancement (**MMEval Score of 86.4%**) versus those from the UCA Yuan et al. (2024) dataset (**MMEval Score of 80.5%**). Using MMEval as a metric, the results demonstrate that the enhancement in the dataset significantly improves the quality and relevance of descriptiveness. This comparison has been conducted only with UCA Yuan et al. (2024) and not CUVA Du et al. (2024) because CUVA has been tested on a completely different dataset, whereas UCA and our method are based on the same dataset.

#### 4.5.2 Crime Localization (When?)

We now provide comparisons between the proposed approach and recent crime localization methods. The methods include TSVLU Sultani et al. (2024), DeepLocalization Rahman et al. (2024), and VideoChatGPT Maaz et al. (2024). Each of these methods employ a distinct approach to identify temporal regions of interest associated with crime events. The TSVLU Sultani et al. (2024) method achieves 46.1% accuracy using the UCA dataset and applies Temporal Sentence Grounding (TSGV) to match crime descriptions with video segments. DeepLocalization Rahman et al. (2024), which focuses on SynDD2, uses change-point detection to locate key events in videos, achieving 51.0% accuracy. VideoChatGPT Maaz et al. (2024), utilizing a custom driving dataset, refines prompt-based extraction for key points, reaching an accuracy of 57.5%. Our proposed method leverages the enhanced UCA dataset and implements a frame-by-frame similarity assessment using CLIP-based feature matching, achieving an accuracy of 63.1%. We also report IoU@0.3, IoU@0.5, and mAP for temporal localization (see Fig. 5). This aligns with standard evaluation protocols and provides a more reliable view of model performance.

#### 4.5.3 Crime Categorization Methods (What?)

These experiments evaluate the performance of the proposed crime categorization method against prior approaches. Traditional baseline methods, such as those by Sultani et al.Sultani et al. (2018) and Majhi et al.Majhi et al. (2021), yield lower mean accuracies of 13.9% and 34.1%, respectively, across broader anomaly categories, without providing class-wise accuracy metrics. The multi-stream approach by Thakare et al. Thakare et al. (2022) demonstrates improvements, achieving a mean accuracy of 69.56% across 13 crime categories. However, it lacks precision in attribute-based categorization, particularly for visually and contextually distinct crimes. Our proposed approach significantly outperforms prior methods in attribute-rich crime categories such as arson (92.38%), shooting (86.5%), and vandalism (88.41%), with an overall mean accuracy of 89.09% in these categories. As shown in Table 4, the proposed model exhibits better performance in arson, shooting, and vandalism, respectively as compared to the multi-stream approach Thakare et al. (2022). These improvements are notable: 36%, 16%, and 32%, respectively across these categories.

## 5 Conclusion

The research highlights substantial improvements achieved by focusing on crime-causing attributes for crime video analysis. By concentrating on causal attribute based crimes, we leveraged their distinct attributes to enhance anomaly descriptions, yielding significant gains in scene quality, localization, and categorization. Even with a limited crime scope, these defined attributes substantially boosted performance. Future work can aim to adapt this method to include other type of events, enhancing the versatility and reach of video-based crime scene analysis and work towards crime prediction.

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

## A  APPENDIX

### A.1  INTRODUCTION

This supplementary document provides additional information to complement the main paper. It includes a link to the enhanced crime attribute-centric annotations dataset, implementation details, and further qualitative results. Section A.2 presents a Dropbox link to the dataset containing the enhanced crime attribute-centric annotations. Section A.3 provides further insights into the dataset creation process. Section B presents additional results and analyses related to our proposed dataset enhancement.

## A.2 DATASET LINK

For review purposes, we have provided an anonymous Dropbox link to our dataset. The folder contains the enhanced crime attribute-centric annotations. Please note that the annotation folder has been anonymized, and any direct or indirect identification of individuals is unintended and does not reflect the authors' true identities. You can access the dataset annotations by clicking on this link (will be given in the camera-ready submission) , which will redirect you to the Dropbox folder.

## A.3 IMPLEMENTATION DETAILS

### A.3.1 DATASET AUGMENTATION

The UCA datasetYuan et al. (2024) is augmented by generating detailed textual annotations for each video frame. For each frame $X_i$, the Deep Multi-Attribute Recognition (DeepMAR)Li et al. (2018) model extracts an attribute vector $\mathbf{A}_i$, encapsulating contextual elements such as objects, actions, or scene details. Using the original UCA label $Y_i$, an initial prompt $P_i$ is generated. This prompt is enriched using a Large Language Model (LLM), which integrates $P_i$ with the extracted attributes $\mathbf{A}_i$ to produce an enhanced prompt $\tilde{P}_i$. The augmented dataset $\tilde{\mathcal{D}} = \{X_i, Y_i, \tilde{P}_i\}_{i=1}^N$ provides granular, attribute-centric annotations, facilitating advanced video understanding tasks such as crime scene explanation. This data augmentation process is summarized in Algorithm 1.

---

**Algorithm 1** Dataset Augmentation with Attribute Annotations

---

1: Initialize UCA dataset $\mathcal{D} = \{X_i, Y_i\}_{i=1}^N$
2: **for** each frame $X_i$ in $\mathcal{D}$ **do**
3:  Extract attribute vector $\mathbf{A}_i = \text{DeepMAR}(X_i)$
4:  Generate initial prompt $P_i$ from UCA labels
5:  Create enhanced prompt $\tilde{P}_i = \text{LLM}(P_i, \mathbf{A}_i)$
6: **end for**
7: Augmented dataset $\tilde{\mathcal{D}} = \{X_i, Y_i, \tilde{P}_i\}_{i=1}^N$

---

## A.4 CRIME CAUSING ATTRIBUTE DETECTION

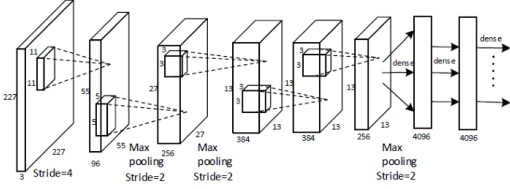

Figure 6: A DeepMAR CNN Architecture used to find crime-causing attributes. Li et al. (2018)

DeepMAR (Deep Multi-Attribute Recognition)Li et al. (2018) employs convolutional neural networks to detect and classify multiple attributes within images, making it particularly effective for analyzing complex crime scene scenarios. In the context of crime-causing attribute detection, the architecture (Figure 6) is designed to extract relevant features from surveillance images and predict the presence of specific attributes, such as weapons, suspicious behavior, or other key indicators associated with criminal activities.

The architecture consists of several key components:

- **Feature Extraction:** The CNN layers in DeepMAR extract critical features from input images, learning both low-level features (e.g., edges, textures) and high-level features (e.g., objects, people, and contextual elements).
- **Multi-Attribute Prediction:** Following feature extraction, DeepMAR simultaneously predicts multiple attributes, such as the presence of weapons, types of activities, or other significant details, which aid in interpreting crime scenes.

- **Fusion Layer:** Some implementations incorporate a fusion layer to integrate extracted features and attribute predictions, offering a holistic understanding of the scene's context and enhancing crime detection capabilities.
- **Training:** The model is trained on a large, annotated dataset of images with labeled attributes. This supervised learning approach enables the network to discern meaningful relationships between visual features and crime-relevant attributes.

By leveraging this architecture, DeepMAR enhances the ability to detect and categorize complex crime-related events, providing a powerful tool for surveillance systems and law enforcement.

### A.4.1 CAN WE TRY WITH OTHER CRIME CATEGORIES ?

These experiments highlight the advantages of the proposed CLIP-based framework and demonstrate how each component—LLM selection, dataset augmentation, attribute-based enhancement, and an improved localization—contributes to superior performance in crime analysis. The effectiveness of the proposed method in harnessing the finer details of attribute-centric crimes such as arson, vandalism, and shooting has been established through rigorous experiments. Our experiments reveal that if the prompts are curated properly, some of the underlying applications will benefit. For example, categorization of certain types of crime becomes more accurate ($\sim 20\% \uparrow$), where causal-attributes are available. However, challenges may arise due to the ambiguity in some attributes. For instance, consider Figure 7, which depicts a robber (left) and a customer (right) inside a convenience store. Both individuals exhibit similar visual features. Without additional contextual information, and absence of causative attributes, the proposed method may not reliably differentiate between criminal and non-criminal behavior. This necessitates integration of a broader contextual or behavioral information to enhance the accuracy further.

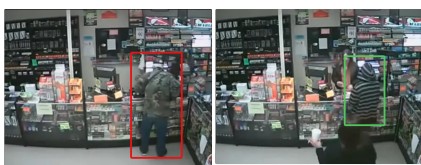

Figure 7: Left image represents robber in a store. Right image represents a normal customer. Can we find unique and definite set of attributes to identify the robber?

## B RESULTS

### B.1 EXAMPLES

Figure 8 represents examples for Arson Crime Category. Figure 9 represents examples for Shooting Crime Category. Figure 10 represents examples for Vandalism Crime Category.

### B.1.1 FALSE POSITIVES

False positives present a significant challenge in anomaly detection, particularly when using LLMs to generate descriptive captions. In the example shown in Figure 11, a frame from the UCA dataset depicts actions such as spray-painting graffiti on a wall and an individual pointing at the camera. While these actions suggest vandalism, the enhanced prompt generated by our method includes specific attributes like "gun" and "spray can." However, the LLM misinterpreted the act of pointing at the camera as a shooting incident, resulting in a false positive. This example underscores the nuanced challenges of aligning detailed descriptions with accurate interpretations. Although our proposed method enhances the granularity of prompts by incorporating crime-causing attributes, it can unintentionally introduce misinterpretations when visual cues are ambiguous. Addressing this issue requires further refinement of prompt generation techniques to balance specificity with contextual accuracy.

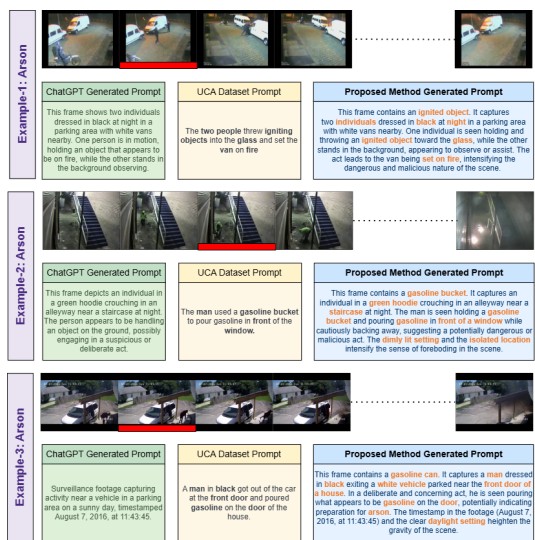

Figure 8: Comparisons of different captioning methods for frames highlighted by the red bar in each example include the initial ChatGPT prompt, the UCA dataset prompt, and the proposed prompt enhancement. These methods are applied to describe arson. The level of detail in each caption significantly impacts the model's accuracy in detecting and interpreting anomalies. Enhanced prompts consistently provide more comprehensive descriptions of anomaly events across all categories, emphasizing critical elements that improve anomaly detection.

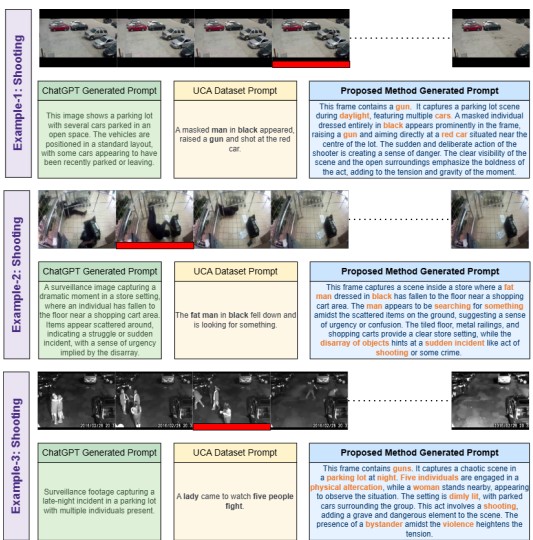

Figure 9: Comparisons of different captioning methods for frames highlighted by the red bar in each example include the initial ChatGPT prompt, the UCA dataset prompt, and the proposed prompt enhancement. These methods are applied to describe shooting incidents. The level of detail in each caption plays a critical role in the model's ability to detect and interpret anomalies accurately. Enhanced prompts consistently provide more detailed and comprehensive descriptions of anomaly events across all categories, emphasizing key elements that enhance anomaly detection.

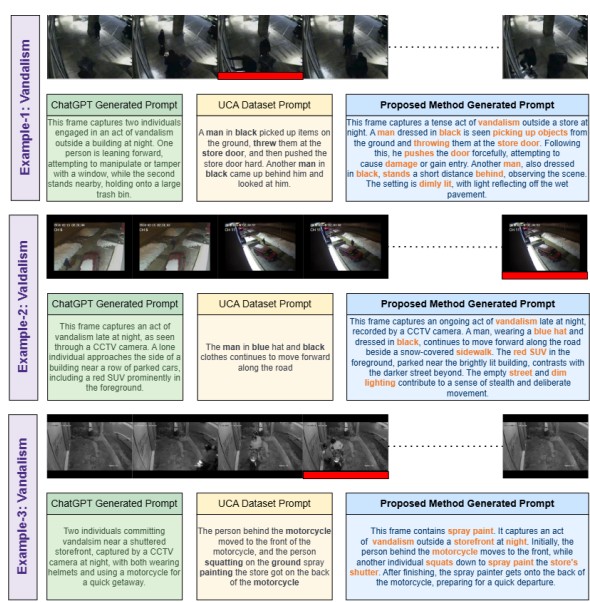

Figure 10: Comparisons of different captioning methods for frames highlighted by the red bar in each example include the initial ChatGPT prompt, the UCA dataset prompt, and the proposed prompt enhancement. These methods are applied to describe vandalism. The level of detail in each caption significantly affects the model's ability to detect and interpret anomalies. Enhanced prompts consistently provide more detailed and comprehensive descriptions of anomaly events across all categories, emphasizing critical elements that enhance anomaly detection.

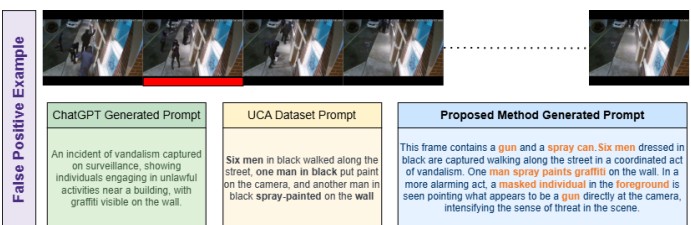

Figure 11: Comparisons of different captioning methods for frames highlighted by the red bar include: (1) the initial ChatGPT prompt, which describes the event as generic vandalism; (2) the UCA dataset prompt, which identifies multiple actions such as spray-painting and covering the camera but omits key elements of the threat; and (3) the proposed enhanced prompt. The enhanced prompt identifies specific attributes, including a gun and a spray can, and describes a more alarming scenario where a masked individual appears to aim a gun at the camera. This level of detail, while comprehensive, led the LLM to misinterpret the action as a shooting incident, highlighting the inherent challenge of balancing specificity with accuracy in anomaly descriptions.

