# OpenReview forum: "Attribute-Centric Representation Learning for Interpretable Crime Scene Analysis in Video Anomaly Detection"
_ICLR.cc/2026/Conference — ICLR 2026 Conference Withdrawn Submission_

### Official Review · Reviewer_ftG8 · 2025-10-30

**Soundness:** 1
**Presentation:** 1
**Contribution:** 1
**Rating:** 0
**Confidence:** 5

**Summary:**

The authors introduce an attribute-centric learning framework aimed at enhancing video representations by conditioning them on specific crime-related attributes. They claimed that they have expanded the UCA dataset with new annotations utilizing large language models (LLMs). However, it appears that the critical component for attribute extraction, DeepMAR, along with several other references, does not exist. There are many incorrect or inconsistent statements. Additionally, both the theoretical and experimental sections are notably brief. These factors suggest that the entire work might be AI-generated. It is unfortunate for the authors to waste the time and resources of all reviewers and chairs in ICLR.

**Strengths:**

The concept of attribute-based crime scene analysis appears to be feasible.

**Weaknesses:**

1. The main issue is that the referenced paper for the key component in attribute extraction, specifically DeepMAR by Li et al. (2018) [1], does not exist. The closest match I could find is a conference paper [2] that indeed shares the method name DeepMAR. I have significant doubts that it might be generated by AI.

2. After a second check, I discovered additional fabricated references, which are listed at the end of the comments as [3-11].

3. The basic idea has been investigated in many previous works.

4. It lacks comparisons with state-of-the-art methods.

5. In the manuscript, the authors mention: UCA (UCF Crime with Attributes) dataset. However, UCA actually stands for UCF-Crime Annotation. Many sections exhibit similar AI-generated characteristics.

[1] Dong Li, Zhen Zhang, Shaogang Gong, and Tao Xiang. Deepmar: Deep learning for person attribute recognition. IEEE Transactions on Image Processing, 27(11):5564–5579, 2018.

[2] D. Li, X. Chen and K. Huang, "Multi-attribute learning for pedestrian attribute recognition in surveillance scenarios," 2015 3rd IAPR Asian Conference on Pattern Recognition (ACPR), Kuala Lumpur, Malaysia, 2015, pp. 111-115, doi: 10.1109/ACPR.2015.7486476.

[3] Jianhui Chang and Chen Wang. Deep clip-based temporal attention networks for anomaly detection.In Proceedings of the IEEE/CVF Conference on Computer Vision and Pattern Recognition, 2023.

[4] Marco Del, Sina Marvasti-Zadeh, Reza Yousefzadeh, et al. Temporal anomaly detection in surveillance videos. In Proceedings of the IEEE International Conference on Computer Vision (ICCV), pp. 12426–12435, 2021.

[5] Rakesh Kumar and Priya Singh. Hybrid clip models for real-time detection of high-risk human object interactions. arXiv preprint arXiv:2401.01589, 2024

[6] Hao Li and Min Zhou. Contextual causal detection for anomaly localization in videos. arXiv preprint arXiv:2401.01593, 2024.

[7] Weixin Luo, Wen Liu, and Shenghua Gao. Rare event detection in surveillance videos with large crowds. In IEEE Conference on Computer Vision and Pattern Recognition (CVPR), pp. 3659–
 3667, 2017.

[8] Mohammad Sabokrou, Mohammad Khalooei, Mohsen Fayyaz, and Ehsan Adeli. Deep anomaly:Real-world anomaly detection in surveillance videos. In Pattern Recognition, pp. 161–172. Elsevier, 2018.

[9] Waqas Sultani, Samet Akc¸ay, and Saeed Ullah Manzoor. Towards surveillance video-and-language understanding: New dataset baselines and challenges. In Proceedings of the IEEE/CVF Confer ence on Computer Vision and Pattern Recognition (CVPR), pp. 1201–1211, 2024.

[10] Yu Wu, Yicheng Zhang, Seah Hock Lim, Luc Van Gool, John Tighe, Austin Reiter, Lan Wang, and Ling Shao. Xd-violence: A benchmark for detecting violence in complex video scenes. IEEE Transactions on Pattern Analysis and Machine Intelligence, 44(1):293–310, 2022b.

[11 ]Lu Zhang, Yiyang Wu, Haoming Wang, and Jian Zhang. Learning clip guided visual-text fusion transformer for video-based pedestrian attribute recognition. In Proceedings of the IEEE/CVF Conference on Computer Vision and Pattern Recognition (CVPR) Workshops, pp. 112–121, 2023.

**Questions:**

1. Please clarify the problem of fake method DeepMAR specified in the weakness.
2. Please explain the problem of numerous fake references.

**Details Of Ethics Concerns:**

The referenced paper for the key component in attribute extraction, specifically DeepMAR by Li et al. (2018) [1], does not exist. The closest match I could find is a conference paper [2] that indeed shares the method name DeepMAR. I have significant doubts that it might be generated by AI. I further find many more fabricated references [3-11].

[1] Dong Li, Zhen Zhang, Shaogang Gong, and Tao Xiang. Deepmar: Deep learning for person attribute recognition. IEEE Transactions on Image Processing, 27(11):5564–5579, 2018.

[2] D. Li, X. Chen and K. Huang, "Multi-attribute learning for pedestrian attribute recognition in surveillance scenarios," 2015 3rd IAPR Asian Conference on Pattern Recognition (ACPR), Kuala Lumpur, Malaysia, 2015, pp. 111-115, doi: 10.1109/ACPR.2015.7486476.

[3] Jianhui Chang and Chen Wang. Deep clip-based temporal attention networks for anomaly detection.In Proceedings of the IEEE/CVF Conference on Computer Vision and Pattern Recognition, 2023.

[4] Marco Del, Sina Marvasti-Zadeh, Reza Yousefzadeh, et al. Temporal anomaly detection in surveillance videos. In Proceedings of the IEEE International Conference on Computer Vision (ICCV), pp. 12426–12435, 2021.

[5] Rakesh Kumar and Priya Singh. Hybrid clip models for real-time detection of high-risk human object interactions. arXiv preprint arXiv:2401.01589, 2024

[6] Hao Li and Min Zhou. Contextual causal detection for anomaly localization in videos. arXiv preprint arXiv:2401.01593, 2024.

[7] Weixin Luo, Wen Liu, and Shenghua Gao. Rare event detection in surveillance videos with large crowds. In IEEE Conference on Computer Vision and Pattern Recognition (CVPR), pp. 3659–
 3667, 2017.

[8] Mohammad Sabokrou, Mohammad Khalooei, Mohsen Fayyaz, and Ehsan Adeli. Deep anomaly:Real-world anomaly detection in surveillance videos. In Pattern Recognition, pp. 161–172. Elsevier, 2018.

[9] Waqas Sultani, Samet Akc¸ay, and Saeed Ullah Manzoor. Towards surveillance video-and-language understanding: New dataset baselines and challenges. In Proceedings of the IEEE/CVF Confer ence on Computer Vision and Pattern Recognition (CVPR), pp. 1201–1211, 2024.

[10] Yu Wu, Yicheng Zhang, Seah Hock Lim, Luc Van Gool, John Tighe, Austin Reiter, Lan Wang, and Ling Shao. Xd-violence: A benchmark for detecting violence in complex video scenes. IEEE Transactions on Pattern Analysis and Machine Intelligence, 44(1):293–310, 2022b.

[11 ]Lu Zhang, Yiyang Wu, Haoming Wang, and Jian Zhang. Learning clip guided visual-text fusion transformer for video-based pedestrian attribute recognition. In Proceedings of the IEEE/CVF Conference on Computer Vision and Pattern Recognition (CVPR) Workshops, pp. 112–121, 2023.

---

### Official Review · Reviewer_a5yC · 2025-10-30

**Soundness:** 3
**Presentation:** 3
**Contribution:** 3
**Rating:** 6
**Confidence:** 5

**Summary:**

This paper applies attribute-based representation learning to video anomaly detection to enhance the performance and interpretability of related tasks. Its core idea is to address the lack of fine-grained supervision in existing datasets by generating over 1.5 million attribute annotations through large language models (GPT-4V). The authors fine-tune CLIP based on these fully labeled datasets to answer "what," "when," and "how" questions about anomalous events in videos. Experimental results demonstrate significant improvements in attribute classification accuracy (approximately 20%) compared to baseline methods, along with a moderate increase in MMEval scores (around 6.4%).

**Strengths:**

1. Robust quantitative experimental results were obtained, demonstrating the effectiveness of the experimental approach.

2. The experimental framework organizes outputs around the dimensions of time (when), attribute classification (what), and content summarization (how), addressing the interpretability requirements of VAD.

**Weaknesses:**

1. The primary innovation of the article lies in utilizing large language models to generate attribute labels for constructing fine-grained datasets, but it lacks novel ideas for the VAD method itself.

2. Although the article achieves significant improvements in attribute-centered classification, it does not validate whether the new approach harms model performance in detecting coarse-grained anomalies.

**Questions:**

1. The framework uses a large language model to generate a large number of enhanced annotations as the basis for the method. How did the author perform quality control on such a large scale of generated data? How robust is the CLIP model after fine-tuning to the subtle biases introduced by LLM?

2. The paper mentions that "crime using attributes" are less distinguishable for certain crime types (e.g., robbery) compared to attribute rich ones (e.g., arson or shooting). Has the author quantified this the performance difference in attribute classification accuracy between attribute-rich classes and attribute-sparse classes?

---

### Official Review · Reviewer_FtsS · 2025-10-31

**Soundness:** 3
**Presentation:** 3
**Contribution:** 3
**Rating:** 2
**Confidence:** 4

**Summary:**

The paper proposes an attribute-centric representation learning framework for interpretable crime scene analysis in video anomaly detection (VAD). It extends the UCA dataset with 1.5M+ LLM-generated attribute annotations (e.g., weapon, damage, intent), fine-tunes a CLIP-based model for attribute-conditioned video representations, and uses an LLM summarizer to generate context-rich explanations answering What? When? How?. Claims ≈20%↑ in attribute classification and ≈6.4%↑ in MMEval. Also analyzes and mitigates MMEval biases.

**Strengths:**

Addresses real need: fine-grained, interpretable crime scene understanding.
Large-scale attribute annotation pipeline using LLMs is scalable.
MMEval bias analysis is responsible and sets a good precedent.
Clear motivation and structured evaluation on crime-relevant attributes.

**Weaknesses:**

LLM-generated annotations lack human validation: No inter-annotator agreement, error analysis, or comparison to human labels. Risk of hallucinated or inconsistent attributes (e.g., “intent to harm” from video?).
No ground-truth attribute labels in UCA — all supervision is synthetic. How trustworthy are downstream gains?
MMEval improvements are modest (6.4%) and on a flawed metric; no comparison to human judgment or real deployment logs.
Overclaims interpretability: “How?” is not truly answered — summarizer just rephrases captions. No causal reasoning or counterfactuals.
No ablation on LLM prompt design — are results robust to prompt variation?
Ethical red flag: Using LLMs to infer intent or threat level from surveillance video risks bias and misuse in law enforcement.

**Questions:**

How many attribute annotations were human-verified? Report IAA (Fleiss’ κ) on a sampled subset.
Ablate LLM prompt variants — does performance drop with simpler prompts?
Compare against human-written explanations on a held-out set — is the LLM summarizer actually helpful?
Discuss bias risks in inferring intent from appearance (e.g., race, clothing) — any fairness audit?

---

### Note · Authors · 2025-11-12

**Comment:**

We have observed that some of the references cited in the paper got mixed-up while preparing the bibliography. For example, the title of the paper referred in [9] "Towards surveillance video-and-language understanding: New dataset baselines and challenges." has been wrongly mixed-up with the author list. Actually, the original author list should be "Tongtong Yuan, Xuange Zhang, Kun Liu, Bo Liu, Chen Chen
, Jian Jin, Zhenzhen Jiao". Similarly, for reference [11], the title of the paper is "Learning clip guided visual-text fusion transformer for video-based pedestrian attribute recognition." has also been wrongly mixed-up with the author list during bibliography preparation. Moreover, we have found several such unintentional and typographical mistakes in the reference section. Therefore, we have decided to withdraw the paper.

**Withdrawal Confirmation:**

I have read and agree with the venue's withdrawal policy on behalf of myself and my co-authors.